# VAST: Value Function Factorization with Variable Agent Sub-Teams

**Thomy Phan**[1]      **Fabian Ritz**[1]      **Lenz Belzner**[2]

**Philipp Altmann**[1]      **Thomas Gabor**[1]      **Claudia Linnhoff-Popien**[1]

[1]LMU Munich
[2]Technische Hochschule Ingolstadt
`thomy.phan@ifi.lmu.de`

## Abstract

Value function factorization (VFF) is a popular approach to cooperative multi-agent reinforcement learning in order to learn local value functions from global rewards. However, state-of-the-art VFF is limited to a handful of agents in most domains. We hypothesize that this is due to the flat factorization scheme, where the VFF operator becomes a performance bottleneck with an increasing number of agents. Therefore, we propose VFF with *variable agent sub-teams (VAST)*. VAST approximates a factorization for sub-teams which can be defined in an arbitrary way and vary over time, e.g., to adapt to different situations. The sub-team values are then linearly decomposed for all sub-team members. Thus, VAST can learn on a more focused and compact input representation of the original VFF operator. We evaluate VAST in three multi-agent domains and show that VAST can significantly outperform state-of-the-art VFF, when the number of agents is sufficiently large.

## 1 Introduction

Many real-world problems can be defined as cooperative *multi-agent system (MAS)*, where multiple autonomous agents collaborate to achieve a common goal like fleet management [20, 21], industry 4.0 [9, 29, 43], or communication networks [26, 51]. *Multi-agent reinforcement learning (MARL)* seems promising to realize such cooperative MAS by learning local policies for each autonomous agent [3, 27, 37, 40]. *Multi-agent credit assignment* is an important challenge, where all agents only observe a single global reward, which makes the deduction of individual agent contributions difficult, especially in large MAS with many agents. This can lead to poor policies, since it is unclear which agent policy needs to adapt to what extent in order to improve global MAS behavior [6, 10, 39].

*Value function factorization (VFF)* via end-to-end deep learning is a popular approach to MARL in order to address the credit assignment problem [31, 32, 36, 39, 44]. A centralized value function is learned from global rewards and factorized into local value functions, which can be used to realize coordinated local policies via multi-armed bandits [32, 40] or local actor-critic learning [30, 38, 45].

Despite the popularity of VFF, most approaches have been only evaluated in domains with a handful of agents. We hypothesize that this is due to the flat factorization scheme of current VFF approaches (Fig. 1a). With an increasing number of agents, the centralized VFF operator becomes a *performance bottleneck*, where it gets difficult to provide sufficiently informative training signal for each agent.

To alleviate this performance bottleneck problem, we propose VFF with *variable agent sub-teams (VAST)*. Instead of directly factorizing a centralized value function for each agent, VAST approximates

35th Conference on Neural Information Processing Systems (NeurIPS 2021).

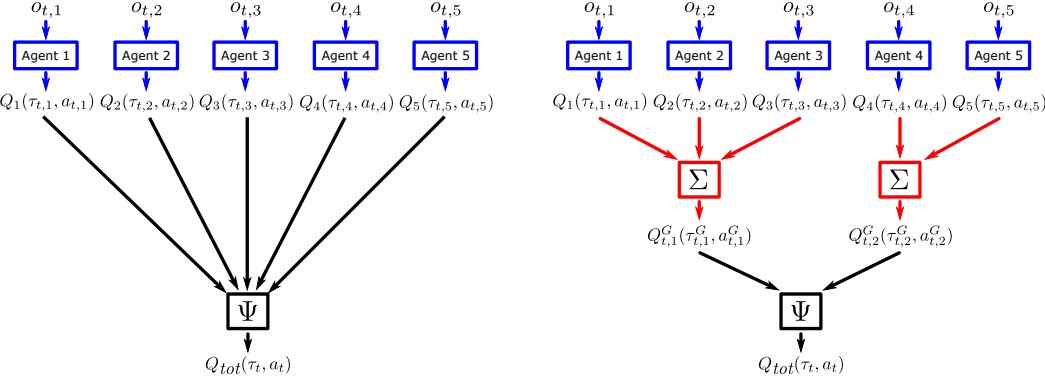

(a) Flat value function factorization for $N = 5$ agents

(b) Factorization for $K = 2$ agent sub-teams

Figure 1: Illustration of different value function factorization schemes using a factorization operator $\Psi$. (a) Flat factorization directly based on local values $Q_i$ per agent $i \in \mathcal{D}$. (b) Proposed factorization based on $K \leq N = |\mathcal{D}|$ sub-team values $Q_{t,k}^G$, which are linearly decomposed into local values $Q_j$ per sub-team member $j \in G_{t,k} \subseteq \mathcal{D}$. Each agent sub-team $G_{t,k}$ is defined by an assignment strategy as explained in Section 4.

a factorization for agent sub-teams which can be defined in an arbitrary way and vary over time, e.g., to adapt to different situations. The sub-team values are then linearly decomposed for all sub-team members as illustrated in Fig. 1b. Therefore, VAST can learn on a more focused and compact input representation of the original VFF operator. Our contributions are as follows:

- We formulate VAST and show that VAST maintains decentralizability like state-of-the-art VFF given any sub-team assignment and depending on the sub-team based VFF operator.

- We propose a meta-gradient approach to optimize sub-team assignments in order to adapt and improve VAST. We also briefly discuss alternative sub-team assignment strategies.

- We empirically evaluate different variants of VAST in three multi-agent domains and show that VAST can significantly outperform flat state-of-the-art VFF approaches by alleviating the performance bottleneck problem, when the number of agents is sufficiently large.

## 2 Background

We model cooperative MAS as partially observable *Markov game* $M = \langle \mathcal{D}, \mathcal{S}, \mathcal{A}, \mathcal{P}, \mathcal{R}, \mathcal{Z}, \Omega \rangle$, where $\mathcal{D} = \{1, ..., N\}$ is a set of agents $i$, $\mathcal{S}$ is a set of states $s_t$ at time step $t$, $\mathcal{A} = \langle \mathcal{A}_i \rangle_{i \in \mathcal{D}}$ is the set of joint actions $a_t = \langle a_{t,i} \rangle_{i \in \mathcal{D}} = \langle a_{t,1}, ..., a_{t,N} \rangle$, $\mathcal{P}(s_{t+1}|s_t, a_t)$ is the transition probability, $r_t = \mathcal{R}(s_t, a_t) \in \mathbb{R}$ is the global reward, $\mathcal{Z}$ is a set of local observations $z_{t,i}$ for each agent $i$, and $\Omega(s_t, a_t) = z_{t+1} = \langle z_{t+1,i} \rangle_{i \in \mathcal{D}} \in \mathcal{Z}^N$ is the subsequent joint observation. Each agent $i$ maintains a local *history* $\tau_{t,i} \in (\mathcal{Z} \times \mathcal{A}_i)^t$ and $\tau_t = \langle \tau_{t,i} \rangle_{i \in \mathcal{D}}$ is the *joint history*. $\pi(a_t|\tau_t) = \prod_{i \in \mathcal{D}} \pi_i(a_{t,i}|\tau_{t,i})$ is the (joint) action probability of *joint policy* $\pi$, where $\pi_i$ is the *local policy* of agent $i$. $\pi$ can be evaluated with a *value function* $Q^\pi(s_t, a_t) = \mathbb{E}_\pi[R_t|s_t, a_t], \forall s_t \in \mathcal{S}, \forall a_t \in \mathcal{A}$, where $R_t = \sum_{c=0}^{\infty} \gamma^c r_{t+c}$ is the *return* with $\gamma \in [0,1)$. The goal is to find an *optimal joint policy* $\pi^* = \langle \pi_i^* \rangle_{i \in \mathcal{D}}$ with $Q^{\pi^*} = Q^* = max_\pi Q^\pi$. If $Q^*$ is known, then $\pi^*$ can be obtained by greedily maximizing $Q^*$.

**Note**: Expressions of the form $\langle e_i \rangle_{i \in \mathcal{I}}$ denote *unordered sets*, where $e_i$ is mapped to exactly one identifier (e.g., an agent) $i \in \mathcal{I}$. Thus, we implicitly assume the order of agents to be irrelevant.

### 2.1 Independent Learning of Value Functions

$Q^*$ can be approximated independently by each agent $i \in \mathcal{D}$ using naive decentralized MARL on $a_{t,i}$ and $\tau_{t,i}$ [10, 18, 40]. These local approximations $Q_i \sim Q^*$ can be used to realize local polices $\pi_i$ for each agent $i$ by using, e.g., multi-armed bandits on $Q_i$ [32, 40] or actor-critic learning with $Q_i$ as critic [10]. *Independent Learning (IL)* offers optimal scalability w.r.t. $N$ but violates the Markov assumption due to non-stationarity caused by simultaneously learning agents [17, 36].

## 2.2 Value Function Factorization

For many problems, training usually takes place in a laboratory or in a simulated environment, where global information is available. State-of-the-art MARL exploits this fact to approximate a centralized value function $Q_{tot} \sim Q^*$, which conditions on joint histories $\tau_t$ and joint actions $a_t$ (and optionally on global states $s_t$). However, $Q_{tot}$ is only required *during training* in order to realize local policies $\pi_i$, which can be used in a decentralized way because they only condition on the local history $\tau_{t,i}$. This paradigm is known as *centralized training and decentralized execution (CTDE)* [10, 32, 36].

$Q_{tot}$ can be factorized into local value functions $\langle Q_i \rangle_{i \in \mathcal{D}}$ via a *VFF operator* $\Psi$ as shown in Fig. 1a:

$$Q_{tot}(\tau_t, a_t) = \Psi(Q_1(\tau_{t,1}, a_{t,1}), ..., Q_N(\tau_{t,N}, a_{t,N})) \tag{1}$$

In practice, $\Psi$ is realized with deep neural networks, such that $\langle Q_i \rangle_{i \in \mathcal{D}}$ can be learned end-to-end via backpropagation by minimizing the mean squared *TD($\lambda$)* (*temporal difference*) error [32, 36, 39]. A VFF operator $\Psi$ is *decentralizable* when satisfying the *IGM (Individual-Global-Max)* such that [36]:

$$argmax_{a_t \in \mathcal{A}} Q_{tot}(\tau_t, a_t) = \langle argmax_{a_{t,i} \in \mathcal{A}_i} Q_i(\tau_{t,i}, a_{t,i}) \rangle_{i \in \mathcal{D}} \tag{2}$$

**VDN (Value Decomposition Networks) [39]** formulates $\Psi_{VDN}$ as linear sum such that $Q_{tot}(\tau_t, a_t) = \Psi_{VDN}(\cdot) = \sum_{i \in \mathcal{D}} Q_i(\tau_{t,i}, a_{t,i})$, which satisfies the IGM for $Q_{tot}$ and $\langle Q_i \rangle_{i \in \mathcal{D}}$ [36].

**QMIX [32]** formulates $\Psi_{QMIX}$ as a nonlinear monotonic combination of $\langle Q_i \rangle_{i \in \mathcal{D}}$ with a mixing network. The mixing network is generated by hypernetworks [12] and has nonnegative weights to satisfy the monotonicity condition $\frac{\delta Q_{tot}}{\delta Q_i} \geq 0, \forall i \in \mathcal{D}$ to maintain consistency w.r.t. the IGM [32, 36]. *Weighted QMIX* further improves QMIX by weighting the transition losses w.r.t. over- and underestimation of $Q_{tot}$ [31]

**QTRAN [36]** avoids the additivity and monotonicity constraints of VDN and QMIX respectively by formulating the more general $\Psi_{QTRAN}$, which aims to satisfy

$$\sum_{i \in \mathcal{D}} Q_i(\tau_{t,i}, a_{t,i}) - Q_{tot}(\tau_t, \mathbf{a_t}) + V_{tot}(\tau_t) = \begin{cases} = 0, \mathbf{a_t} = \overline{\mathbf{a_t}} \\ \geq 0, \mathbf{a_t} \neq \overline{\mathbf{a_t}} \end{cases} \tag{3}$$

where $\overline{\mathbf{a_t}} = \langle \overline{a_{t,i}} \rangle_{i \in \mathcal{D}} = \langle \overline{a_{t,1}}, ..., \overline{a_{t,N}} \rangle$ with $\overline{a_{t,i}} = argmax_{a_{t,i} \in \mathcal{A}_i} Q_i(\tau_{t,i}, a_{t,i})$ and $V_{tot}(\tau_t) = max_{a_t \in \mathcal{A}} Q_{tot}(\tau_t, a_t) - \sum_{i \in \mathcal{D}} Q_i(\tau_{t,i}, \overline{a_{t,i}})$, in order to be consistent w.r.t. the IGM.

## 3 Related Work

MARL is a long-standing research area with rapid progress towards complex domains [3, 11, 40, 42]. Most state-of-the-art approaches are based on CTDE to learn $Q_{tot}$ for actor-critic learning [10, 22] or VFF [31, 32, 36, 39, 44]. VFF approaches like VDN, QMIX, and QTRAN use a flat factorization scheme, where $Q_{tot}$ is directly factorized into $\langle Q_i \rangle_{i \in \mathcal{D}}$ as illustrated in Fig. 1a. We introduce a *hierarchical* VFF approach based on *agent sub-teams* which can vary over time, e.g., to adapt to different situations. With that, we can improve performance in large MAS, where flat VFF approaches could fail due to $\Psi$ becoming a performance bottleneck.

Prior work on hierarchical MARL has mainly focused on temporal abstraction, where the MAS attempts to solve tasks based on temporal subgoals or roles [24, 49]. We focus on VFF applied to agent sub-teams, which can be regarded as an *abstraction of agents*. These abstractions or sub-teams can *vary over time*, depending on the sub-team assignment strategy which may be chosen *arbitrarily*.

Approaches based on coordination graphs enrich VFF with agent relationship information and focus on pairwise interactions of agents to learn local value functions [1, 19]. The maximization of $Q_{tot}$ scales at least quadratically w.r.t. $N$ based on the graph structure. Our approach simplifies VFF via *agent abstraction* and maximizes $Q_{tot}$ with *linear* complexity. The abstraction is based on *variable* agent sub-teams, which are not restricted to pairwise interactions.

There is some prior work on sub-team assignments and agent-based hierarchization in MAS: The relationship between coordination, complexity, and performance depending on predefined organizational MAS structures was studied in [4, 8, 15, 34, 35]. [20] proposed a contextual MARL framework for

fleet management, where the spatial environment is partitioned into fixed cells with locally assigned rewards and the number of agents per cell can vary over time. [16] proposed an attention-based mechanism for self-interested MAS to focus on different contextual information per agent in order to approximate $Q_i$. Mean field MARL was introduced in [50], where $Q_i$ is learned based on the mean field approximation of the joint action of all neighbor agents, where the definition of "neighborhood" is domain dependent. Different clustering approaches for agents, communication messages, etc. w.r.t. some similarity criteria have been proposed in [5, 25, 46]. Our approach addresses the *performance bottleneck problem* of flat VFF approaches. It can be used with *any sub-team assignment strategy* like random assignments, clustering, or meta-learning to structure the MAS *dynamically* while satisfying the IGM for $Q_{tot}$ and $\langle Q_i \rangle_{i \in \mathcal{D}}$ like flat state-of-the-art VFF approaches [32, 36, 39]. Unlike [16, 20, 50], our approach does not depend on predefined local rewards per agent or region but automatically approximates local value functions $Q_i$ from *global rewards* via sub-team based VFF.

# 4 Value Function Factorization with Variable Agent Sub-Teams (VAST)

## 4.1 Variable Agent Sub-Teams

We now introduce VFF with *variable agent sub-teams (VAST)*. Given a *sub-team ratio* $\eta \in [\frac{1}{N}; 1]$, VAST divides the set of agents $\mathcal{D}$ into $K = \lceil \eta N \rceil \leq N$ agent *sub-teams* $G_{t,k} \in \mathcal{G}_t$ of *division* $\mathcal{G}_t = \langle G_{t,1}, ..., G_{t,K} \rangle$ at every time step $t$. Each agent $i \in \mathcal{D}$ is assigned to exactly one sub-team $G_{t,k}$ by a *sub-team assignment strategy* $\mathcal{X}$ with distribution $\mathcal{X}(k|i, \tau_{t,i}, s_t), k \in \{1, ..., K\}$ such that $G_{t,k} \subseteq \mathcal{D}$, $G_{t,k} \cap G_{t,k'} = \emptyset$ if $k \neq k'$, and $\mathcal{D} = \bigcup_{k=1}^{K} G_{t,k}$. Each sub-team $G_{t,k}$ can be regarded as temporary *agent abstraction* which selects *sub-team actions* $a_{t,k}^G = \langle a_{t,j} \rangle_{j \in G_{t,k}}$ based on all *sub-team members* $j \in G_{t,k}$. The value function $Q_{t,k}^G$ of $G_{t,k}$ is computed via $\Psi_{VDN}$ on $\langle Q_j \rangle_{j \in G_{t,k}}$:

$$Q_{t,k}^G(\tau_{t,k}^G, a_{t,k}^G) = \Psi_{VDN}(\cdot) = \sum_{j \in G_{t,k}} Q_j(\tau_{t,j}, a_{t,j}) \tag{4}$$

---

**Algorithm 1** Variable Agent Sub-Teams

1: **procedure** *VAST*$(M, \Psi, \mathcal{X}, \eta)$
2:     Initialize parameters of $\Psi$, $\mathcal{X}$, $\langle Q_i \rangle_{i \in \mathcal{D}}$
3:     $K \leftarrow \lceil \eta N \rceil$
4:     **for** episode $x \leftarrow 1, T$ **do**
5:         Sample $s_1$, observe $z_1$
6:         **for** time step $t$ **do**
7:             $a_t \sim \pi(a_t|\tau_t)$
8:             $r_t, z_{t+1} \leftarrow \mathcal{R}(s_t, a_t), \Omega(s_t, a_t)$
9:             $s_{t+1} \sim \mathcal{P}(s_{t+1}|s_t, a_t)$
10:           **for** sub-team $k \leftarrow 1, K$ **do**
11:               $G_{t,k} \leftarrow \{\}$
12:           **for** agent $i \in \mathcal{D}$ **do**
13:               $k \sim \mathcal{X}(k|i, \tau_{t,i}, s_t)$
14:               $G_{t,k} \leftarrow G_{t,k} \cup \{i\}$
15:           $\mathcal{G}_t \leftarrow \langle G_{t,1}, ..., G_{t,K} \rangle$
16:           $Q_{t,k}^G(\tau_{t,k}^G, a_{t,k}^G) \leftarrow Eq.4, \forall G_{t,k}$
17:           Update $\Psi$, $\langle Q_i \rangle_{i \in \mathcal{D}}$ with $TD(\lambda)$
18:           Update $\mathcal{X}$ (e.g., Eq. 6)    ▷ optional

---

Despite the simplified approximation of $Q_{t,k}^G$ in Eq. 4, $\Psi_{VDN}$ has two important advantages over nonlinear variants like $\Psi_{QMIX}$ and $\Psi_{QTRAN}$, which would also satisfy the IGM for $Q_{t,k}^G$ and $\langle Q_j \rangle_{j \in G_{t,k}}$: First, the sum of $\Psi_{VDN}$ has no fixed input dimension, thus sub-team sizes may vary over time, e.g., to adapt to different situations. Second, $\Psi_{VDN}$ does not introduce new tunable hyperparameters, thus being more efficient to use. Therefore, we defer nonlinear approximations of $Q_{t,k}^G$ to future work.

$Q_{tot}$ is approximated from $\langle Q_{t,k}^G \rangle_{G_{t,k} \in \mathcal{G}_t}$ using a VFF operator $\Psi$ according to Eq. 1:

$$Q_{tot}(\tau_t, a_t) = \Psi(Q_{t,1}^G(\tau_{t,1}^G, a_{t,1}^G), ..., Q_{t,K}^G(\tau_{t,K}^G, a_{t,K}^G)) \tag{5}$$

where $K = \lceil \eta N \rceil$ specifies the input dimension of $\Psi$. The computation hierarchy of $Q_{tot}$ based on VAST according to Eq. 4 and 5 is depicted in Fig. 1b. With that hierarchy, $\langle Q_i \rangle_{i \in \mathcal{D}}$ can be learned end-to-end, e.g., via backpropagation by updating $\Psi$ w.r.t. the mean squared $TD(\lambda)$ error.

VAST is formulated in Algorithm 1, where $M$ is the MAS, $\Psi$ is an IGM preserving *VFF operator* like $\Psi_{VDN}$, $\Psi_{QMIX}$, or $\Psi_{QTRAN}$ as listed in Section 2.2 to approximate $Q_{tot}$ from $\langle Q_{t,k}^G \rangle_{G_{t,k} \in \mathcal{G}_t}$, $\mathcal{X}$ is a sub-team assignment strategy, and $\eta \in [\frac{1}{N}; 1]$ is the sub-team ratio.

Table 1: Characteristics of different sub-team assignment strategies $\mathcal{X}$. The worst case complexity indicates the computational overhead per time step $t$ and agent $i$, when invoking $\mathcal{X}$ (line 13 in Algorithm 1) or updating $\mathcal{X}$ (line 18 in Algorithm 1) if all other parameters (e.g., $\eta$) are constant.

| Approach | Description | Worst case complexity | Domain knowledge |
|---|---|---|---|
| $\mathcal{X}_{Random}$ | Random assignment with $\mathcal{X}(k\|i, \tau_{t,i}, s_t) = \frac{1}{K}$ | $\mathcal{O}(1)$ | None |
| $\mathcal{X}_{Fixed}$ | Fixed assignment with deterministic $\mathcal{X}$ | $\mathcal{O}(1)$ | Agent IDs |
| $\mathcal{X}_{Spatial}$ | Spatial clustering of agents to specify $\mathcal{X}$ | $\mathcal{O}(N^C), C > 1$ | Spatial information |
| $\mathcal{X}_{MetaGrad}$ | Meta-gradient learning of $\mathcal{X}(k\|i, \tau_{t,i}, s_t)$ (Eq. 6) | $\mathcal{O}(N)$ | Optional |

$\eta$ specifies the degree of input space compression of $\Psi$. The smaller $\eta$, the more compact the input representation of $\Psi$. In the extreme case of $\eta = \frac{1}{N} \Rightarrow K = 1$, the factorization reduces to $Q_{tot}(\tau_t, a_t) = \Psi(\Psi_{VDN}(\cdot)) = \Psi(\sum_{i \in \mathcal{D}} Q_i(\tau_{t,i}, a_{t,i}))$. Larger values of $\eta$ enable more exploration of the input space of $\Psi$ but at the cost of more compute, which increases linearly w.r.t. $\eta$. Furthermore, we suggest $\frac{1}{N} < \eta \ll 1$ for large $N$ to alleviate the original performance bottleneck problem of $\Psi$.

To show that VAST maintains decentralizability by satisfying the IGM for $Q_{tot}$ and $\langle Q_i \rangle_{i \in \mathcal{D}}$ for an *arbitrary* sub-team assignment strategy $\mathcal{X}$, we formulate and prove Theorem 1:

**Theorem 1.** *Given a MAS $M = \langle \mathcal{D}, \mathcal{S}, \mathcal{A}, \mathcal{P}, \mathcal{R}, \mathcal{Z}, \Omega \rangle$ at time step $t$, where each agent $i \in \mathcal{D}$ with local value function $Q_i$ is assigned to exactly one sub-team $G_{t,k} \in \mathcal{G}_t$ for sub-team based VFF according to Eq. 4: If the IGM is satisfied for a factorization of the centralized value function $Q_{tot}$ into sub-team value functions $\langle Q_{t,k}^G \rangle_{G_{t,k} \in \mathcal{G}_t}$ via a VFF operator $\Psi$ according to Eq. 5, then the IGM is also satisfied for $Q_{tot}$ and $\langle Q_i \rangle_{i \in \mathcal{D}}$ for each agent $i \in \mathcal{D} = G_{t,1} \cup ... \cup G_{t,K}$.*

*Proof.* The factorization of $Q_{tot}$ into $\langle Q_{t,k}^G \rangle_{G_{t,k} \in \mathcal{G}_t}$ via $\Psi$ satisfies the IGM. Thus, the maximization of all $Q_{t,k}^G$ maximizes $Q_{tot}$ such that $\overline{\mathbf{a}}_\mathbf{t} = \langle \overline{\mathbf{a}}_{\mathbf{t,k}}^\mathbf{G} \rangle_{G_{t,k} \in \mathcal{G}_t}$, where $\overline{\mathbf{a}}_\mathbf{t} = argmax_{a_t \in \mathcal{A}} Q_{tot}(\tau_t, a_t)$ and $\overline{\mathbf{a}}_{\mathbf{t,k}}^\mathbf{G} = argmax_{a_{t,k}^G \in \langle \mathcal{A}_i \rangle_{i \in G_{t,k}}} Q_{t,k}^G(\tau_{t,k}^G, a_{t,k}^G)$. The factorization of $Q_{t,k}^G$ into $\langle Q_i \rangle_{i \in G_{t,k}}$ via $\Psi_{VDN}$ (Eq. 4) also satisfies the IGM such that $\overline{\mathbf{a}}_{\mathbf{t,k}}^\mathbf{G} = \langle \overline{a}_{t,i} \rangle_{i \in G_{t,k}}$, where $\overline{a}_{t,i} = argmax_{a_{t,i} \in \mathcal{A}_i} Q_i(\tau_{t,i}, a_{t,i})$:

$$\overline{\mathbf{a}}_\mathbf{t} \overset{\Psi}{=} \langle \overline{\mathbf{a}}_{\mathbf{t,k}}^\mathbf{G} \rangle_{G_{t,k} \in \mathcal{G}_t} \overset{\Psi_{VDN}, Eq.4}{=} \langle \langle \overline{a}_{t,i} \rangle_{i \in G_{t,k}} \rangle_{G_{t,k} \in \mathcal{G}_t} \overset{\mathcal{D}=G_{t,1} \cup ... \cup G_{t,K}}{=} \langle \overline{a}_{t,i} \rangle_{i \in \mathcal{D}}$$

Therefore, the set of greedy local actions of all agents $\langle \overline{a}_{t,i} \rangle_{i \in \mathcal{D}} = \langle \overline{a}_{t,1}, ..., \overline{a}_{t,N} \rangle = \overline{\mathbf{a}}_\mathbf{t}$ maximizes $Q_{tot}$ for *any* sub-team assignment according to the IGM in Eq. 2 which is ensured by the whole hierarchy of VAST as illustrated in Fig. 1b. $\square$

### 4.2 Sub-Team Assignment Strategies

According to Theorem 1, VAST ensures IGM consistency w.r.t. arbitrary sub-team assignments. To enable adaptation to different situations, the assignment of sub-teams per time step can be regarded as a decision making problem by using a *meta-policy* $\mathcal{X}$ to select sub-team assignments conditioned on states and agent information. $\mathcal{X}$ can be optimized w.r.t. some *meta-objective* $J$ to further improve performance of VAST. $J$ represents a high-level objective like the return $R_t$ as defined in Section 2 or some domain specific goal which can be evaluated in an outer loop, e.g., line 18 in Algorithm 1.

We propose the *meta-gradient* based assignment strategy or meta-policy $\mathcal{X}_{MetaGrad}$ inspired by [48]. $\mathcal{X}_{MetaGrad}$ is approximated with parameter vector $\theta$, which is automatically optimized via gradient ascent on the meta-objective $J(\theta)$ w.r.t. to the following estimated gradient:

$$g = \hat{A}(k, i, \tau_{t,i}, s_t) \nabla_\theta log \mathcal{X}_{MetaGrad}(k|i, \tau_{t,i}, s_t) \tag{6}$$

where $\hat{A}(k, i, \tau_{t,i}, s_t) = \hat{Q}(k, i, \tau_{t,i}, s_t) - \hat{V}(i, \tau_{t,i}, s_t)$ is the advantage of $k$ for sub-team $G_{t,k} \in \mathcal{G}_t$ given $i$, $\tau_{t,i}$, and $s_t$. $\hat{Q}$ estimates the (expected) performance when selecting $k$ given $i$, $\tau_{t,i}$, and $s_t$. $\hat{V}$ represents a baseline function, which can depend on $i$, $\tau_{t,i}$, and $s_t$, for variance reduction. The concrete definitions of $\hat{Q}$ and $\hat{V}$ are based on $J(\theta)$, which should correlate with the original target of $Q_{tot}$ and can optionally integrate domain knowledge. In this paper, we estimate $\hat{A}(k, i, \tau_{t,i}, s_t)$ by

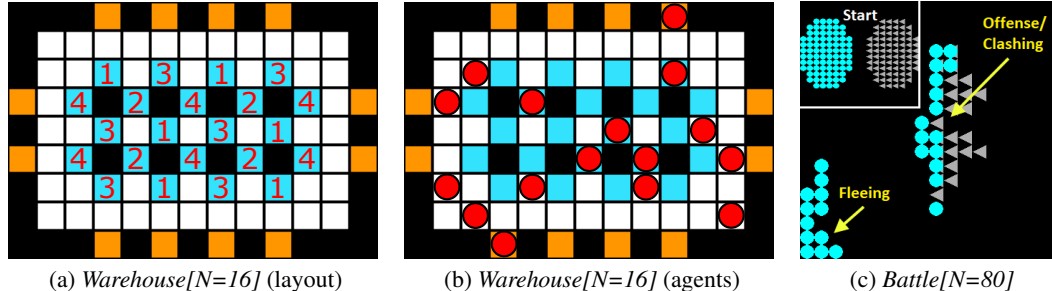

| (a) *Warehouse[N=16]* (layout) | (b) *Warehouse[N=16]* (agents) | (c) *Battle[N=80]* |

Figure 2: Illustration of the *Warehouse[N]* and the *Battle[N]* domain. (a) Work stations (orange cells) and drop off locations (cyan cells) in *Warehouse[16]*. (b) All agents (red circles) need to pick up orders of 5 items $b_w \in \{1, 2, 3, 4\}$ at the work stations and deliver them to the corresponding drop off locations according to (a) while avoiding stalling and collisions with other agents. (c) An army of learning agents (cyan circles) has to fight another army of opponent agents (gray triangles).

setting $\hat{Q}(k, i, \tau_{t,i}, s_t) = R_t$ to the return and $\hat{V}(i, \tau_{t,i}, s_t) = \sum_{a_{t,i} \in \mathcal{A}_i} \pi_i(a_{t,i}|\tau_{t,i}) Q_i(\tau_{t,i}, a_{t,i})$ to the expected local value of agent $i$ with value function $Q_i$ and local policy $\pi_i$ to avoid any additional domain dependencies. For simplicity, we propose on-policy training of $\mathcal{X}_{MetaGrad}$. Further extensions to, e.g., off-policy training or enhanced exploration are left for future work.

Table 1 lists some alternative sub-team assignment strategies for comparison: $\mathcal{X}_{Random}$ assigns each agent to a random sub-team at every time step, while $\mathcal{X}_{Fixed}$ permanently assigns each agent to a particular sub-team based on its ID. $\mathcal{X}_{Spatial}$ uses spatial information like coordinates to cluster agents in order to form sub-teams. $\mathcal{X}_{MetaGrad}$ optimizes sub-team assignments w.r.t. some meta-objective to adapt to different situations and to further improve VAST. $\mathcal{X}_{Spatial}$ and $\mathcal{X}_{MetaGrad}$ introduce additional computational overhead per time step and agent depending on $N$.

## 5 Experimental Setup

### 5.1 Evaluation Domains

To assess the scalability of VAST in comparison with flat VFF approaches, we focus on domains that can be easily scaled up to large numbers of agents $N$. Since common benchmarks like StarCraft are currently limited to $N < 30$ agents [33], we defer an evaluation on these benchmarks to future versions which support significantly more agents.

**Warehouse[N]** is a grid-world environment inspired by [8, 9, 43] and illustrated in Fig. 2a-b, where $N$ agents or robots have to pick up randomly generated *orders* of 5 *items* $b_w \in \{1, 2, 3, 4\}, w \in \{1, ..., 5\}$ at *work stations* and deliver each item $b_w$ to its corresponding *drop off location*, where the drop off number according to Fig. 2a matches $b_w$. All agents start at random work stations. After delivering all items of an order, the agent can return to any work station for a new order. All agents have a $5 \times 5$ field of view and are able to pick up and drop off their items if possible, move north, south, west, east, or do nothing. Agents cannot share positions or occupy obstacle cells. Delivered items and completed orders are rewarded with +1. Collisions with other agents are penalized with -0.5. At every time step, there is a time penalty of -0.01. An episode ends after 50 time steps.

**Battle[N]** is a grid-world environment inspired by [52] and illustrated in Fig. 2c, where an army of $N$ learning agents has to fight another army of $N$ *opponent* agents, which randomly move towards and attack all learning agents in sight. Each agent $i$ initially has 3 *health points* ($HP_i$), which are recovered by 0.01 at each time step when $0 < HP_i < 3$. An agent $i$ is *dead* or *killed* when $HP_i = 0$. All agents have a $7 \times 7$ field of view and are able to move north, south, west, east, do nothing, or attack one opponent if occupying the same cell, resulting in the attacked opponent's loss of one health point. Successful attacks and kills are rewarded with +1. Attacking a cell without any opponent is penalized with -0.1 and being hit or killed by the opponent is penalized with -0.5. An episode ends after 100 time steps or when all agents of an army have been killed.

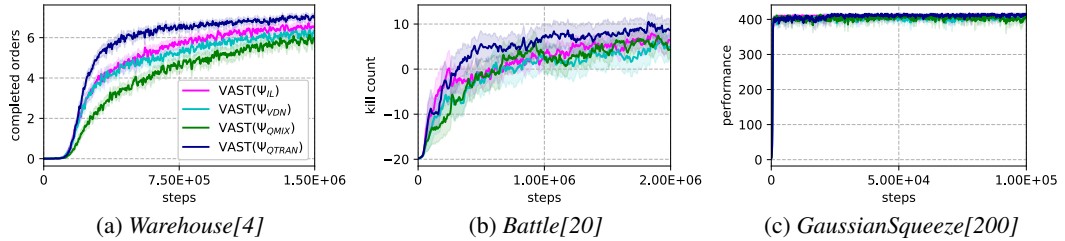

(a) *Warehouse[4]*        (b) *Battle[20]*        (c) *GaussianSqueeze[200]*

Figure 3: Average training progress of VAST with $\Psi \in \{\Psi_{IL}, \Psi_{VDN}, \Psi_{QMIX}, \Psi_{QTRAN}\}$, $\mathcal{X}_{MetaGrad}$, and $\eta = \frac{1}{2}$. Shaded areas show the 95% confidence interval. Legend in (a) applies to all plots.

**GaussianSqueeze[N]** is a single-step multi-agent resource allocation problem introduced in [14], where $N$ agents have to coordinate their actions $a_{t,i} \in \mathcal{A}_i = \{0, ..., 9\}$ to find the most efficient *allocation* $\zeta = \sum_{i=1}^{N} a_{t,i}$. The *system performance* is defined by $GS(\zeta) = \zeta e^{\frac{-(\zeta - \mu)^2}{\sigma^2}}$ and $\mu$ and $\sigma$ are domain dependent parameters, which we set to $\mu = 400$ and $\sigma = 200$.

## 5.2 Learning Algorithms and Training

We implemented IL, QMIX, and QTRAN as baselines. For VAST, we use the notation VAST($\Psi, \mathcal{X}, \eta$) with VFF operator $\Psi \in \{\Psi_{IL}, \Psi_{VDN}, \Psi_{QMIX}, \Psi_{QTRAN}\}$ ($\Psi_{IL}$ approximates $Q_{t,k}^G = Q_{tot}$ independently), sub-team assignment strategy $\mathcal{X} \in \{\mathcal{X}_{Random}, \mathcal{X}_{Fixed}, \mathcal{X}_{Spatial}, \mathcal{X}_{MetaGrad}\}$, and sub-team ratio $\eta \in \{\frac{1}{4}, \frac{1}{2}\}$ (Algorithm 1). We chose $\frac{1}{4}$ as minimum value for $\eta$ because it is the smallest possible value for *Warehouse[4]*. Since value-based algorithms are highly sensitive w.r.t. the exploration decay schedule, we use $Q_i$ as critic for local actor-critic learning to realize $\pi_i$ in order to evaluate all approaches on a common basis [30, 38, 45]. $\mathcal{X}_{Fixed}$ assigns each agent $i$ to sub-team $G_{t,k}$ with $k = i \pmod{K} + 1$. $\mathcal{X}_{Spatial}$ uses k-means clustering on the agents' $(x, y)$-positions in *Warehouse[N]* and *Battle[N]* with $\frac{K}{2}$ centroids. If not stated otherwise, we assume the following defaults: $\Psi = \Psi_{QTRAN}$, $\mathcal{X} = \mathcal{X}_{MetaGrad}$.

We performed 30 training runs for each MARL algorithm and report the domain-specific performance, i.e., the number of *completed orders* in *Warehouse[N]*, the *kill count* in *Battle[N]* (kills by opponent agents are counted negatively), and the *system performance* in *GaussianSqueeze[N]* respectively.

Further details on the training setup and the experiments are specified in Appendix A.1 and A.2.

## 6 Results

### 6.1 Comparison of Value Function Factorization Operators for VAST

We ran VAST with different VFF operators $\Psi \in \{\Psi_{IL}, \Psi_{VDN}, \Psi_{QMIX}, \Psi_{QTRAN}\}$, $\mathcal{X} = \mathcal{X}_{MetaGrad}$, and $\eta = \frac{1}{2}$ in *Warehouse[4]*, *Battle[20]*, and *GaussianSqueeze[200]*. The results are shown in Fig. 3. All variants show steady learning progress in all domains. VAST($\Psi_{QTRAN}$) performs best in *Warehouse[4]* and *Battle[20]*. In *GaussianSqueeze[200]*, all variants perform equally well.

### 6.2 State-of-the-Art Comparison

We ran VAST with different sub-team ratios $\eta \in \{\frac{1}{4}, \frac{1}{2}\}$, $\Psi = \Psi_{QTRAN}$, and $\mathcal{X} = \mathcal{X}_{MetaGrad}$ in *Warehouse[4]*, *Battle[20]*, and *GaussianSqueeze[200]* as well as in larger instances, i.e., *Warehouse[16]*, *Battle[80]*, and *GaussianSqueeze[800]* (medium instances are shown in Appendix A.3.1) to compare the performance with QMIX, QTRAN, and IL as shown in Fig. 4. In *Warehouse[4]*, QTRAN makes slightly faster progress than VAST($\eta = \frac{1}{2}$). However in *Warehouse[16]*, both VAST variants outperform all baselines, which perform poorly. In *Battle[20]*, both VAST variants slightly outperform QMIX and QTRAN, but they perform significantly better in *Battle[80]*. In *GaussianSqueeze[200]*, all CTDE approaches perform equally well, but both VAST variants clearly outperform all baselines in *GaussianSqueeze[800]*. VAST($\eta = \frac{1}{2}$) initially improves faster than VAST($\eta = \frac{1}{4}$) in most domains but in *Warehouse[16]* and *GaussianSqueeze[800]*, VAST($\eta = \frac{1}{4}$) surpasses VAST($\eta = \frac{1}{2}$) over time.

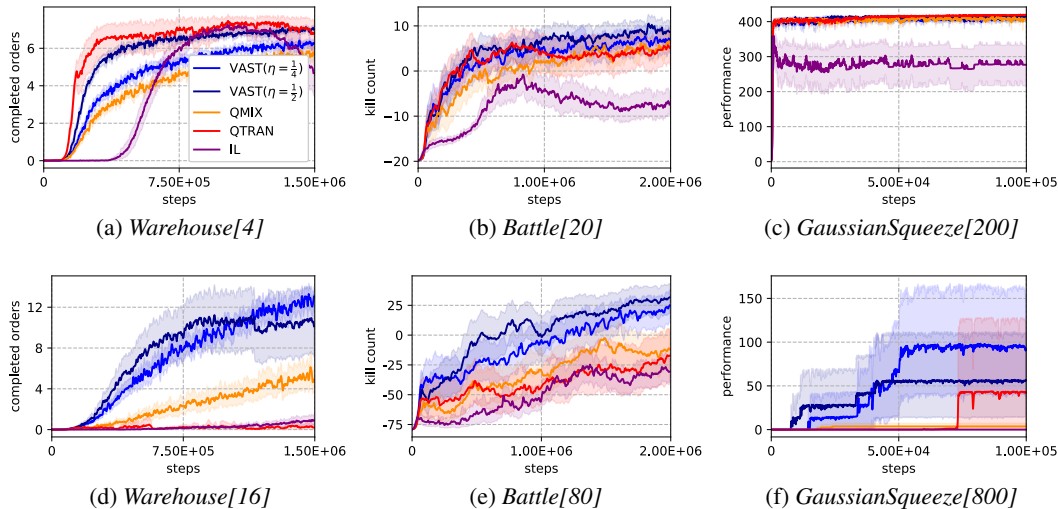

Figure 4: Average training progress of VAST with $\eta \in \{\frac{1}{4}, \frac{1}{2}\}$, $\Psi_{QTRAN}$, and $\mathcal{X}_{MetaGrad}$ as well as QMIX, QTRAN, and IL. Shaded areas show the 95% confidence interval. Legend in (a) applies to all plots. The full results of the state-of-the-art comparison are shown in Fig. 8 in Appendix A.3.1.

## 6.3 Comparison of Sub-Team Assignment Strategies for VAST

We ran VAST with different $\mathcal{X} \in \{\mathcal{X}_{Random}, \mathcal{X}_{Fixed}, \mathcal{X}_{Spatial}, \mathcal{X}_{MetaGrad}\}$, $\Psi = \Psi_{QTRAN}$, and $\eta = \frac{1}{4}$ in *Warehouse[16]*, *Battle[80]*, and *GaussianSqueeze[800]* to compare the performance with the respective best baselines from Fig. 4 in Section 6.2. $\mathcal{X}_{Spatial}$ was not tested in *GaussianSqueeze[800]*, due to the lack of spatial information. The results are shown in Fig. 5. VAST($\mathcal{X}_{MetaGrad}$) performs best in all domains. In *Battle[80]*, VAST($\mathcal{X}_{Fixed}$) is competitive to VAST($\mathcal{X}_{MetaGrad}$) while VAST($\mathcal{X}_{Random}$) and VAST($\mathcal{X}_{Spatial}$) are competitive to the best baseline. In *Warehouse[16]* and *GaussianSqueeze[800]*, all VAST variants clearly outperform the respective best baselines.

We further examined the generated sub-teams of $\mathcal{X}_{MetaGrad}$ and $\mathcal{X}_{Spatial}$ at different stages in *Battle[80]* by visualizing all agents of the same sub-team with the same color in Fig. 6. In the early stage (Fig. 6a), $\mathcal{X}_{MetaGrad}$ generates a cyan sub-team for agents that are rather far away from the opponent army and a red sub-team which is rather close to it (with some prediction noise). In the middle stage (Fig. 6b), a yellow sub-team emerges, when both armies clash, which disappears later (Fig 6c), when the opponent army is significantly decimated, thus reverting back to the cyan and red sub-teams depending on the agent positions. $\mathcal{X}_{Spatial}$ simply groups agents according to their spatial distances to each other with no obvious relation to the danger of the current situation as shown in Fig. 6d-f.

Since the opponent army follows an offensive strategy, most learned policies adopted a defensive strategy, where all agents group and defend themselves together like in Fig. 6. However, in some cases, VAST learned a "splitting" strategy, where the army splits into a *fleeing part* to reduce overall deaths and an *offensive part* that clashes with the opponent army to increase the kill count as shown in Fig. 2c. The generated sub-teams of the splitting strategy are shown in Fig. 10 in Appendix A.3.3.

Although $\eta = \frac{1}{4}$ would theoretically enable $\mathcal{X}_{MetaGrad}$ to assign agents to $K = 20$ different sub-teams, only a few sub-teams are actually active or non-empty as indicated in Fig. 6a-c and Fig. 10a-c in Appendix A.3.3. $\mathcal{X}_{MetaGrad}$ learns to focus on particular sub-teams in specific situations, thus only activates certain sub-teams to adapt on demand, e.g., as indicated by the yellow sub-team in Fig. 6b.

## 7 Discussion

We proposed VAST to approximate value function factorizations for agent sub-teams which can be defined in an arbitrary way and vary over time, e.g., to adapt to different situations. The sub-team values are then linearly decomposed for all sub-team members. VAST learns on a more focused

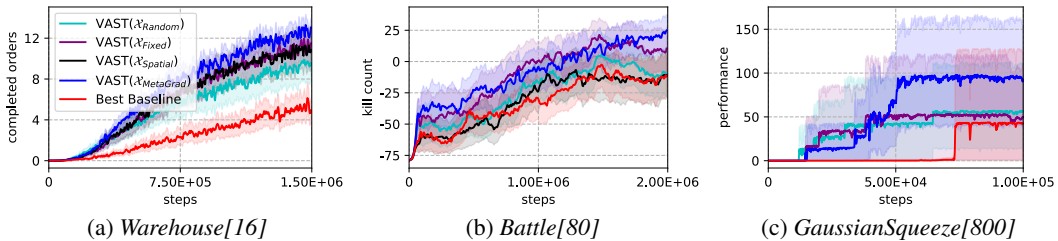

Figure 5: Average training progress of VAST with $\mathcal{X} \in \{\mathcal{X}_{Random}, \mathcal{X}_{Fixed}, \mathcal{X}_{Spatial}, \mathcal{X}_{MetaGrad}\}, \Psi_{QTRAN}$, and $\eta = \frac{1}{4}$ as well as the respective best baselines from Fig. 4 in Section 6.2. Shaded areas show the 95% confidence interval. Legend in (a) applies to all plots.

and compact input representation of the original VFF operator, thus being able to better address the multi-agent credit assignment problem in larger MAS than flat state-of-the-art VFF approaches.

Our experiments show that VAST is able to learn with different VFF operators $\Psi$ to improve performance in domains, where flat VFF approaches could fail to learn meaningful factorizations. This is clearly shown for QTRAN and VAST($\Psi_{QTRAN}$) in the large MAS instances *Warehouse[16]*, *Battle[80]*, and *GaussianSqueeze[800]* in Fig. 4d-f, where VAST($\Psi_{QTRAN}$) significantly outperforms QTRAN. The poor scalability of QTRAN confirms the findings of previous works [23, 31, 44] and shows that VAST is an effective approach to significantly improve scalability. The difference between IL and VAST($\Psi_{IL}$) can already be seen in the small MAS instances *Warehouse[4]*, *Battle[20]* and *GaussianSqueeze[200]*, where IL lacks stability in Fig. 4a-c, while VAST($\Psi_{IL}$) improves steadily in these domains as shown in Fig. 3a-c. VAST can significantly outperform flat state-of-the-art VFF approaches like QMIX and QTRAN by alleviating the performance bottleneck problem, when the number of agents is sufficiently large as shown in Fig. 4d-f and Fig. 5.

VAST achieves competitive or superior performance with arbitrary sub-team assignment strategies $\mathcal{X}$ as shown in Fig. 5, which is supported by Theorem 1. $\mathcal{X}_{MetaGrad}$ is an adaptive approach, which optimizes sub-team assignments to further improve VAST. In *Battle[80]*, $\mathcal{X}_{MetaGrad}$ is able to meaningfully structure the MAS according to different situations, which might be more beneficial for VAST than just relying on simple domain dependent features like agent IDs [2, 10] or spatial positions [20, 50] as shown in Fig. 6 and 10 in Appendix A.3.3. However, $\mathcal{X}_{MetaGrad}$ introduces additional computational overhead, which scales linearly per agent w.r.t. $N$ as stated in Table 1. Furthermore, the learning quality strongly depends on the meta-objective definition of $\mathcal{X}_{MetaGrad}$ [48].

In Fig. 4d and Fig. 4f, VAST($\eta = \frac{1}{2}$) itself suffers from the performance bottleneck (but to a much lesser extent than flat VFF), where VAST($\eta = \frac{1}{4}$) improves more stably and surpasses VAST($\eta = \frac{1}{2}$) over time. However, VAST($\eta = \frac{1}{2}$) is superior in *Battle[80]* and performs better in the early training stages in *Warehouse[16]*. In Fig. 9 in Appendix A.3.2, this is indicated by more exploration through a higher sub-team division diversity. Due to the potential performance bottleneck of $\Psi$ and the computational scaling w.r.t. $\eta$, we recommend VAST for large MAS with $\frac{1}{N} < \eta \ll 1$ for high efficiency and performance. A self-tuning mechanism for $\eta$ would be interesting for future work.

The linear approximation of sub-team values ensures flexibility w.r.t. sub-teams and makes additional hierarchization of sub-teams obsolete, due to the associative property of additions (e.g., $(a+b)+(c+d) = a+b+c+d$). However, this might be too restrictive for some domains. Using nonlinear variants like recurrent neural networks [7, 13] or transformers [28, 41] could further improve flexibility and performance but requires more compute and yields higher complexity due to more hyperparameters. Thus, we defer an investigation on more flexible VAST schemes to future work.

## 8    Potential Negative Societal Impacts

The goal of our work is to realize autonomous systems to solve complex tasks at large scale in a distributed way as motivated in Section 1. To focus completely on the potential effects of our work, we refer to [47] for a general overview regarding societal implications of deep reinforcement learning.

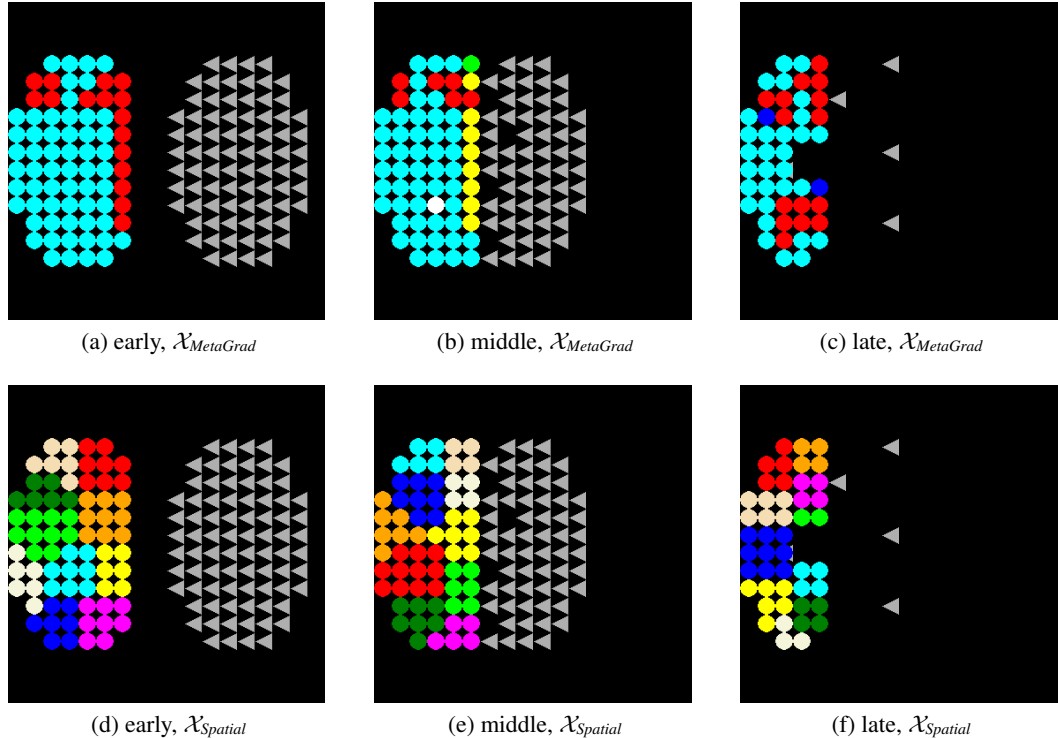

| (a) early, $\mathcal{X}_{MetaGrad}$ | (b) middle, $\mathcal{X}_{MetaGrad}$ | (c) late, $\mathcal{X}_{MetaGrad}$ |
| (d) early, $\mathcal{X}_{Spatial}$ | (e) middle, $\mathcal{X}_{Spatial}$ | (f) late, $\mathcal{X}_{Spatial}$ |

Figure 6: Visualizations of the generated sub-teams of $\mathcal{X}_{MetaGrad}$ with $\eta = \frac{1}{4}$ and $\mathcal{X}_{Spatial}$ with k-means clustering using 10 centroids at different stages (early, middle, late) in *Battle[80]* after training. All learning agents (round circles) of the same sub-team have the same color.

VAST is a CTDE approach with a centralized training regime to realize decentralized policies. These policies have a common objective which might include bias of a central authority and can cause harm to opposing parties, e.g., via discrimination or misleading information. Since we assume VAST to be trained in a laboratory or in a simulation, the trained system might exhibit unsafe behavior when being deployed into the real world due to poor generalization, e.g., by causing traffic accidents. Depending on the choice of $\Psi$, $\mathcal{X}$, and $\eta$, some computational overhead is added to the original VFF approach, which can be significant when scaling up. The generated sub-teams can potentially be used to evaluate and categorize living beings w.r.t. some assignment strategy $\mathcal{X}$ and objective as shown in Fig. 6 and 10 in Appendix A.3.3, which could lead to misuse or discrimination of particular groups.

As experimentally shown in the *Battle[N]* domain, VAST can be misused for real combat, e.g., in autonomous weapon systems to realize coordinated and distributed strategies as demonstrated in Fig. 6 and 10 in Appendix A.3.3. MAS trained with VAST can be assumed to be resilient w.r.t. single agent failures (e.g., agent deaths in *Battle[N]*), which can make human intervention (e.g., shutting down the MAS by disabling single agents) difficult. Behavioral changes of single agents due to updates, failures, or malicious attacks could lead to unexpected emergent effects like adaptive sub-team reorganizations, which can cause, e.g., traffic jams, outages of critical infrastructure, or directly harm to others, depending on the quality of the learned policies and the common objective.

## Acknowledgments and Funding Disclosure

We thank the members of the Mobile and Distributed Systems Group at LMU Munich. This project has received funding from the Bavarian Ministry of Economic Affairs, Regional Development, and Energy under the project "Innovationszentrum Mobiles Internet".

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
