# A Appendix

## A.1 Technical Details

### A.1.1 Neural Network Architectures and Policy Approximation

We used deep neural networks to implement $\pi_i$ and $Q_i$ for each agent $i$. The neural networks are updated every 10 episodes (Appendix A.1.2) using ADAM with a learning rate of 0.001.

Since *Warehouse[N]* and *Battle[N]* are gridworlds, states and observations are encoded as multi-channel image as proposed in [1, 3]. In *GaussianSqueeze[N]*, the state is a zero vector, and the observations are one-hot vectors of dimension $N$ with the number one being at the $i$th position.

We implemented all neural networks as multilayer perceptron (MLP) and flattened the inputs before feeding them into the networks. $\pi_i$ and $Q_i$ have two hidden layers of 64 units with ELU activation. The output of $\pi_i$ has $|\mathcal{A}_i|$ units with softmax activation. The output of $Q_i$ has $|\mathcal{A}_i|$ linear units. The centralized $\Psi$-networks for learning $Q_{tot}$ have two hidden layers of 128 units with ELU activation and one linear output unit. The neural network for $\mathcal{X}_{MetaGrad}$ has two hidden layers of 128 units with ELU activation and $K = \lceil \eta N \rceil$ output units with softmax activation.

The architectures of $\pi_i$ and $Q_i$ are based on architectures of previous MARL work [2, 5] to avoid exhaustive tuning. The architecture of the centralized networks for $Q_{tot}$ and $\mathcal{X}_{MetaGrad}$ is based on [6]. The layers are wider because of the larger input dimension with state $s_t$ and joint action $a_t$.

We approximated $\pi_i$ using local actor-critic learning on $Q_i$ as proposed in [4, 6, 7] w.r.t. the estimated gradient $g = A(\tau_{t,i}, a_{t,i}) \nabla_\theta log \pi_i(a_{t,i}|\tau_{t,i})$, where $A(\tau_{t,i}, a_{t,i}) = R_t - V(\tau_{t,i})$ is the advantage with return $R_t$ as defined in Section 2 and baseline $V(\tau_{t,i}) = \sum_{a_{t,i} \in \mathcal{A}_i} \pi_i(a_{t,i}|\tau_{t,i}) Q_i(\tau_{t,i}, a_{t,i})$.

### A.1.2 Hyperparameters

All common hyperparameters used by all MARL approaches in the experiments as reported in Section 5 and 6 in the paper as well as in Appendix A.3 are listed in Table 2. The final values were chosen based on a coarse grid search on *Warehouse[4]* and *Battle[20]* for QMIX, QTRAN, and VAST (with default parameters according to Section 5.2) with 5 runs each. We directly adopted the final values in Table 2 for all other MARL approaches without further tuning. For $\mathcal{X}_{Spatial}$, we used the k-means implementation of `sklearn 0.24.2` with the default settings without further tuning. All penalty weights of QTRAN-base in $\Psi_{QTRAN}$ were set to 1 as proposed in [5] without further tuning.

Table 2: Common hyperparameters and their respective final values used by all algorithms evaluated in the paper. We also list the numbers that have been tried during development of the paper.

| Hyperparameter | Final Value | Numbers/Range | Description |
|---|---|---|---|
| Learning rate | 0.001 | {0.001} | We used the default value of ADAM in `torch 1.7.0` without further tuning. |
| Clip norm parameter | 1 | {1,$\infty$} | Gradient clipping parameter. Using a clip norm of 1 leads to better performance than disabling it with $\infty$. |
| Discount factor $\gamma$ | 0.95 | {0.9, 0.95, 0.99} | Discount factor for the return $R_t$. Any value $\geq 0.95$ would have been sufficient. |
| Trace decay $\lambda$ | 1 | {0, 1} | Used for TD($\lambda$) learning of $\Psi$. $\lambda$ was set to $\lambda = 1$ to simplify training and to reduce computation time. |
| Local history length | 1 | {1, 5, 10} | The history length was set to 1 to reduce computation time because the other values did not significantly improve performance. |

A training run consists of $T = 3000$ episodes in *Warehouse[N]*, $T = 2000$ episodes in *Battle[N]*, and $T = 10,000$ episodes in *GaussianSqueeze[N]*. The run length $T$ was determined by the convergence

of VAST in the small MAS instances, i.e., *Warehouse[4]*, *Battle[20]*, and *GaussianSqueeze[200]*. After every 10th episode, we performed a gradient update on the neural networks according to the final hyperparameter values in Table 2 and ran 10 test episodes whose average results are shown in all plots in Section 6 and Appendix A.3.

### A.1.3 Computing Infrastructure, Resources, and Total Amount of Compute

All training runs in the experiments and for hyperparameter tuning were performed in parallel on a computing cluster of fifteen x86_64 GNU/Linux (Ubuntu 18.04.5 LTS) machines with i7-8700 @ 3.2GHz CPU (8 cores) and 64 GB RAM using Slurm-WLM 19.05.5. The amount of compute depends on the MARL algorithm and on the domain. Table 3 gives an overview of the estimated average runtimes per training run of VAST with $\Psi = \Psi_{QTRAN}$ and $\mathcal{X} = \mathcal{X}_{MetaGrad}$, QMIX, QTRAN, and IL for each domain in Fig. 4 and 8. Note that the average runtimes are only rough measurements to estimate the total amount of compute as provided below.

Table 3: Estimated average runtimes of VAST, QMIX, QTRAN, and IL for each domain.

| Domain | Algorithm | Average Runtime per Run |
|---|---|---|
| *Warehouse[4]* | VAST($\eta = \frac{1}{4}$) | ~2.5 hours |
| | VAST($\eta = \frac{1}{2}$) | ~2.5 hours |
| | QMIX, QTRAN, IL | ~2.5 hours |
| *Warehouse[8]* | VAST($\eta = \frac{1}{4}$) | ~5 hours |
| | VAST($\eta = \frac{1}{2}$) | ~5 hours |
| | QMIX, QTRAN, IL | ~5 hours |
| *Warehouse[16]* | VAST($\eta = \frac{1}{4}$) | ~14 hours |
| | VAST($\eta = \frac{1}{2}$) | ~14 hours |
| | QMIX, QTRAN, IL | ~14 hours |
| *Battle[20]* | VAST($\eta = \frac{1}{4}$) | ~6.5 hours |
| | VAST($\eta = \frac{1}{2}$) | ~6.5 hours |
| | QMIX, QTRAN, IL | ~4.9 hours |
| *Battle[40]* | VAST($\eta = \frac{1}{4}$) | ~22.5 hours |
| | VAST($\eta = \frac{1}{2}$) | ~23.2 hours |
| | QMIX, QTRAN, IL | ~15 hours |
| *Battle[80]* | VAST($\eta = \frac{1}{4}$) | ~2 days |
| | VAST($\eta = \frac{1}{2}$) | ~2.5 days |
| | QMIX, QTRAN, IL | ~1.5 days |
| *GaussianSqueeze[200]* | VAST($\eta = \frac{1}{4}$) | ~7.5 hours |
| | VAST($\eta = \frac{1}{2}$) | ~9 hours |
| | QMIX, QTRAN, IL | ~3 hours |
| *GaussianSqueeze[400]* | VAST($\eta = \frac{1}{4}$) | ~0.8 days |
| | VAST($\eta = \frac{1}{2}$) | ~1.3 days |
| | QMIX, QTRAN, IL | ~0.3 days |
| *GaussianSqueeze[800]* | VAST($\eta = \frac{1}{4}$) | ~2.7 days |
| | VAST($\eta = \frac{1}{2}$) | ~4.7 days |
| | QMIX, QTRAN, IL | ~0.6 days |

The runtime was not significantly different for other VFF operators $\Psi$ that are evaluated in Fig. 3 in Section 6.1. The runtime of using $\mathcal{X}_{Random}$ or $\mathcal{X}_{Fixed}$ instead of $\mathcal{X}_{MetaGrad}$ as sub-team assignment strategy $\mathcal{X}$ as evaluated in Fig. 5 in Section 6.3 was similar to the runtime of VAST($\eta = \frac{1}{4}$). However, the runtime was roughly doubled compared to VAST($\eta = \frac{1}{4}$) when using $\mathcal{X}_{Spatial}$ instead of $\mathcal{X}_{MetaGrad}$.

Our initial experiments (implementation, debugging, hyperparameter tuning, etc.) required about 5000 CPU hours of compute. The experiments presented in Section 6 and Appendix A.3 required about 40,000 CPU hours of compute. Thus, our work required about **45,000 CPU hours** of total compute.

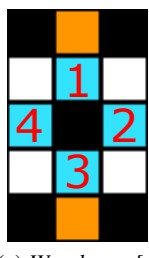  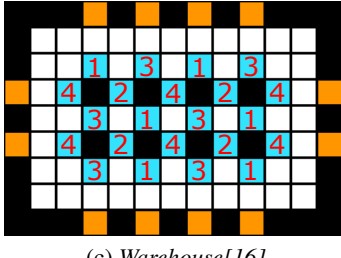

| (a) *Warehouse[4]* | (b) *Warehouse[8]* | (c) *Warehouse[16]* |

Figure 7: Layouts used in the experiments in Section 6 and Appendix A.3.1 for *Warehouse[N]* with work stations (orange cells), drop off locations (cyan cells), and obstacles (black cells).

## A.2 Domain Details

### A.2.1 Warehouse[N]

We experimented with different instances of *Warehouse[N]* with $N \in \{4, 8, 16\}$. The layouts used for the respective instances are shown in Fig. 7. The goal is to complete as many orders with +6 reward (due to 5 randomly assigned items and a completion bonus) as possible while avoiding collisions with other agents which are penalized with -0.5 per "attempt" to occupy the same position as another agent. With an increasing number of agents, the chance of agent collisions significantly increases, thus we scaled the layout sizes accordingly to keep the domain instances solvable for MARL algorithms.

### A.2.2 Battle[N]

We experimented with different instances of *Battle[N]* with $N \in \{20, 40, 80\}$ with a grid world shape of $10 \times 10$, $14 \times 14$, and $18 \times 18$ respectively. All agents are theoretically able to share the same positions and are only able to attack opponent agents, when sharing the same position with them. If multiple opponent agents occupy the same position, then a random opponent is picked for the attack. Due to these rules, it is recommended to group together in order to attack simultaneously. E.g., 3 simultaneously attacking agents can kill a single opponent within just one turn without loosing any health points. However a single agent requires at least 4 turns for a kill, due to the recovery of 0.01 health point per turn, while getting hit itself by the opponent agent.

## A.3 Additional Results

### A.3.1 Full State-of-the-Art Comparison

We ran VAST with different sub-team ratios $\eta \in \{\frac{1}{4}, \frac{1}{2}\}$, $\Psi = \Psi_{QTRAN}$, and $\mathcal{X} = \mathcal{X}_{MetaGrad}$ in the small MAS instances *Warehouse[4]*, *Battle[20]*, and *GaussianSqueeze[200]*. We also experimented with medium instances, i.e., *Warehouse[8]*, *Battle[40]*, and *GaussianSqueeze[400]* as well as larger instances, i.e., *Warehouse[16]*, *Battle[80]*, and *GaussianSqueeze[800]* to compare the performance with QMIX, QTRAN, and IL as shown in Fig. 8.

In *Warehouse[4]*, QTRAN makes slightly faster progress than VAST($\eta = \frac{1}{2}$). In *Warehouse[8]*, VAST($\eta = \frac{1}{2}$) performs best and VAST($\eta = \frac{1}{4}$) is only outperformed by QTRAN. In *Warehouse[16]*, both VAST variants outperform all baselines, which perform poorly. In *Battle[20]* and *Battle[40]*, both VAST variants slightly outperform QMIX and QTRAN, but they perform significantly better in *Battle[80]*. In *GaussianSqueeze[200]*, all CTDE approaches perform equally well, but both VAST variants outperform all baselines in *GaussianSqueeze[400]* and *GaussianSqueeze[800]*. VAST($\eta = \frac{1}{2}$) initially improves faster than VAST($\eta = \frac{1}{4}$) in most domains but in *Warehouse[16]* and *GaussianSqueeze[800]*, VAST($\eta = \frac{1}{4}$) surpasses VAST($\eta = \frac{1}{2}$) over time.

Both VAST variants seem to perform especially well in the medium and large MAS instances, where VAST($\eta = \frac{1}{2}$) tends to perform better in the medium instances, while VAST($\eta = \frac{1}{4}$) performs best in *Warehouse[16]* and *GaussianSqueeze[800]*.

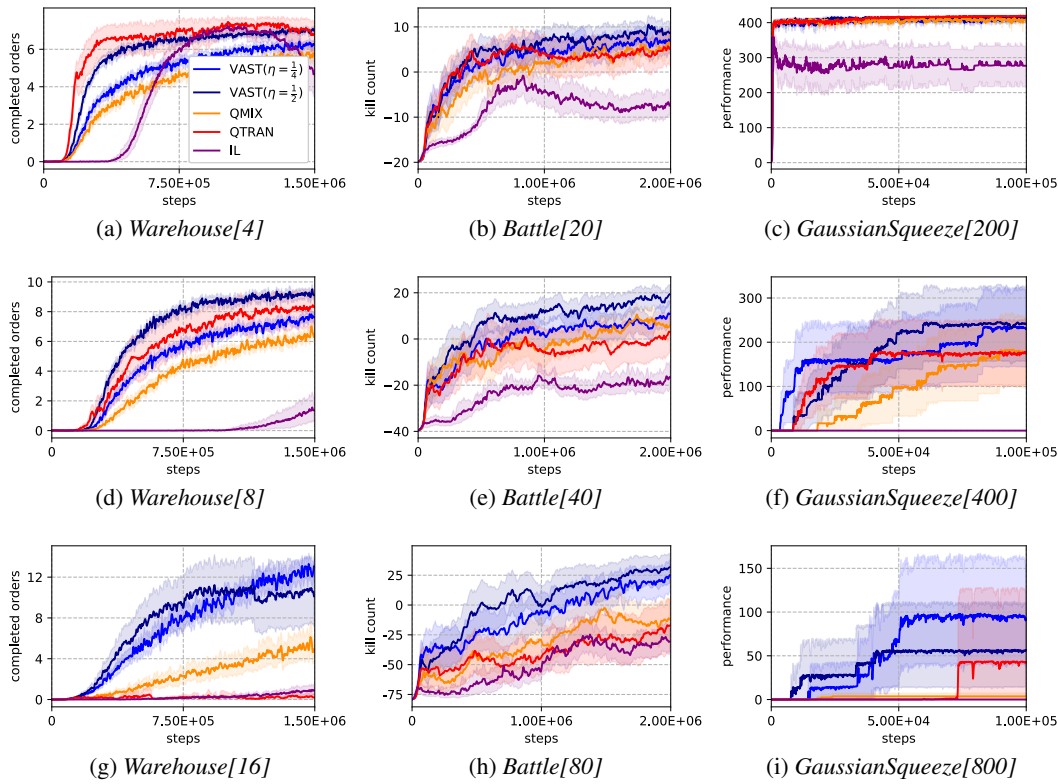

Figure 8: Average training progress of VAST with $\eta \in \{\frac{1}{4}, \frac{1}{2}\}$, $\Psi_{QTRAN}$, and $\mathcal{X}_{MetaGrad}$ as well as QMIX, QTRAN, and IL. Shaded areas show the 95% confidence interval. Legend in (a) applies to all plots.

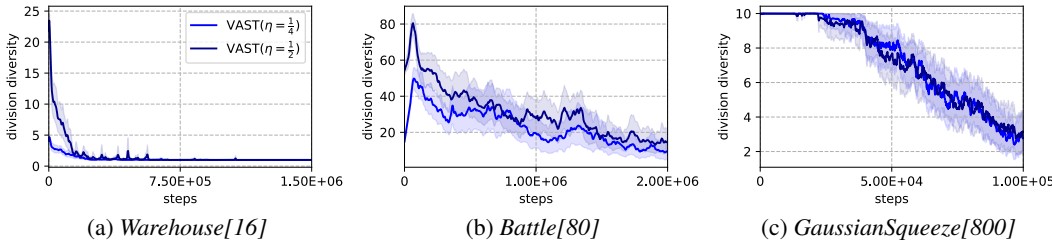

Figure 9: Average progress of the division diversity per episode using $\Psi_{QTRAN}$ and $\mathcal{X}_{MetaGrad}$ during training. The division diversity in *GaussianSqueeze[800]* is determined by 10 single-step episodes. Shaded areas show the 95% confidence interval. Legend in (a) applies to all plots.

### A.3.2 Sub-Team Division Diversity

We evaluated the diversity of sub-team divisions $\mathcal{G}_t$ to examine exploration and the number of active outputs (i.e., non-empty sub-teams) in $\mathcal{X}_{MetaGrad}$ over time for $\eta \in \{\frac{1}{4}, \frac{1}{2}\}$. We define the *division diversity* by the number of non-empty sub-team assignments for all time steps per episode:

$$d_{\mathcal{G}_t} = |\{\mathcal{K}_t | \forall episode\ time\ step\ t\}| \tag{1}$$

where $\mathcal{K}_t = \{k | G_{t,k} \in \mathcal{G}_t \wedge G_{t,k} \neq \emptyset\}$. The results for *Warehouse[16]*, *Battle[80]*, and *GaussianSqueeze[800]* are shown in Fig. 9. In *Warehouse[16]*, both variants converge to an average division diversity of $d_{\mathcal{G}_t} = 1$, where VAST($\eta = \frac{1}{2}$) converges slower and occasionally peaks out. In *Battle[80]*, VAST($\eta = \frac{1}{4}$) reaches an average diversity of $d_{\mathcal{G}_t} \approx 10$, while VAST($\eta = \frac{1}{2}$) reaches an average diversity of $d_{\mathcal{G}_t} \approx 17$. The division diversity in *GaussianSqueeze[800]* is determined by 10 single-step episodes, where both variants progress similarly to an average diversity of $d_{\mathcal{G}_t} \approx 3$.

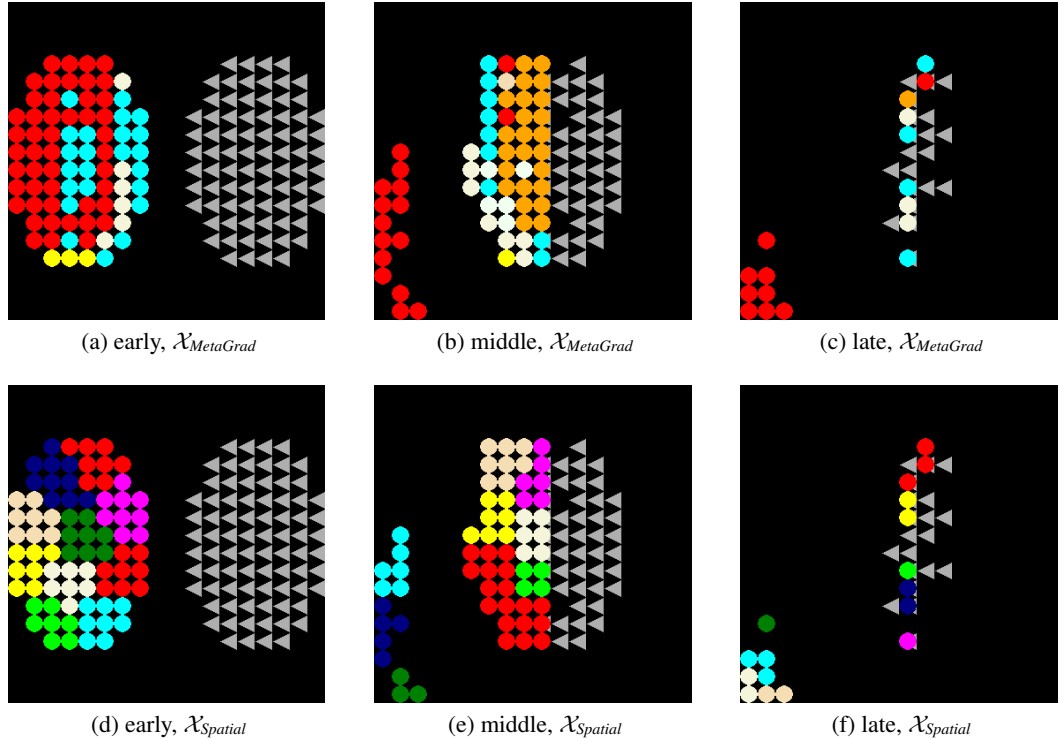

|                          |                          |                          |
|:------------------------:|:------------------------:|:------------------------:|
| (a) early, $\mathcal{X}_{MetaGrad}$ | (b) middle, $\mathcal{X}_{MetaGrad}$ | (c) late, $\mathcal{X}_{MetaGrad}$ |
| (d) early, $\mathcal{X}_{Spatial}$ | (e) middle, $\mathcal{X}_{Spatial}$ | (f) late, $\mathcal{X}_{Spatial}$ |

Figure 10: Visualizations of the generated sub-teams of $\mathcal{X}_{MetaGrad}$ with $\eta = \frac{1}{4}$ and $\mathcal{X}_{Spatial}$ with k-means clustering using 10 centroids at different stages (early, middle, late) in *Battle[80]* after training. All agents of the same sub-team have the same color.

$\mathcal{X}_{MetaGrad}$ automatically learns to adapt in all domains and shrink its number of active or non-empty sub-teams on demand with any sub-team ratio $\eta$. Larger values of $\eta$ can enable more initial exploration as indicated by the higher division diversity of $\mathcal{X}_{MetaGrad}$ with $\eta = \frac{1}{2}$ in the early training stages in *Warehouse[16]* and in *Battle[80]*. The exploration effect is also indicated in Fig. 8, where VAST($\eta = \frac{1}{2}$) initially improves faster than VAST($\eta = \frac{1}{4}$) in *Warehouse[16]* and *Battle[80]*.

### A.3.3   "Splitting" Strategy in Battle[80]

In Fig. 10, we visualize the generated sub-teams of the "splitting" strategy described in Section 6.3 by using $\mathcal{X}_{MetaGrad}$ and $\mathcal{X}_{Spatial}$ at different stages in *Battle[80]*. All learning agents of the same sub-team are circles with the same color. The gray triangles represent opponent agents.

In the early stage (Fig. 10a), $\mathcal{X}_{MetaGrad}$ generates a red sub-team for agents that are rather far away from the opponent army and a cyan/white sub-team which is rather close to it (with some prediction noise). In the middle stage (Fig. 10b), the learning agent army splits into a *fleeing part* as red sub-team and an *offensive part* with an orange sub-team that directly clashes with the opponent army and the cyan/white sub-team backing up the orange one. The offensive part and the opponent army decimate each other, while the red sub-team flees and hides in the bottom left corner (Fig. 10c). $\mathcal{X}_{Spatial}$ simply groups agents according to their spatial distances to each other with no obvious relation to the danger of the current situation or the splitted parts of the learning agent army as shown in Fig. 10d-f.