# OpenReview forum: "VAST: Value Function Factorization with Variable Agent Sub-Teams"
_NeurIPS.cc/2021/Conference — NeurIPS 2021 Poster_

### Official Review · Reviewer_t76M · 2021-07-01

**Rating:** 5
**Confidence:** 4

**Summary:**

This method proposes a hierarchical decomposition for value function factorization methods which enables significantly greater scalability in terms of the number of agents.

**Limitations And Societal Impact:**

The authors could spend some time explaining why the approach does not scale to more complex domains (e.g. StarCraft).

**Main Review:**

# Strengths
* The paper successfully addresses a significant shortcoming of a popular class of algorithms for deep MARL.
* The scalability of the method with respect to agent quantities is particularly commendable.

# Weaknesses
* Section 4.2, on the meta-gradient method for learning $\chi$, should be significantly expanded. Whereas the rest of the method is relatively straightforward, this part is a bit more subtle, and it would be good to read more on the motivation for and methodology behind this choice. In fact, the approach to learning $\chi$ as described in this paper looks more like a straightforward policy gradient approach, rather than the cited meta-gradient method. The cited method is concerned with meta-learning a set of parameters that directly influence the objective function of the policy/value function; whereas, this paper is concerned with learning the parameters of $\chi$, which can just be seen as additional parameters of the value function.
It's also not clear to me why $\chi$ cannot simply be learned in an end-to-end manner where it is implemented, for example, as a differentiable sampling procedure via Gumbel-Softmax.
* The authors do not evaluate on the StarCraft Multi-Agent Challenge [1], the standard benchmark for value function factorization-based cooperative MARL methods, nor do they compare to the most recent state-of-the-art in this field (QPLEX [2]). This omission suggests a lack of scalability to more complex non-gridworld domains.

## Minor Points
* Using $\mathcal{D}$ to represent the set of agents is slightly confusing, as it is typically used to denote a replay buffer or dataset.
* The methods section is a bit dense with respect to notation.

# Questions/Comments
* A recent approach, REFIL [3], proposes a method for extending QMIX to variable-sized teams via attention mechanisms as part of their approach. This may be worth looking into for future extensions of this work, such that the sub-teams may use a more expressive operator than VDN for their factorization scheme.
* Lines 207-209 "Since value-based algorithms are highly sensitive w.r.t. the exploration decay schedule, we completely omit exploration tuning and use $Q_i$ as critic for local actor-critic learning to realize $\pi_i$ instead " - this connection isn't quite clear to me. Actor-critic methods do not implicitly alleviate problems with exploration, they just deal with them differently (e.g. entropy penalty on the policies).

# Overall Thoughts
I like the main idea and motivation behind this paper, as scalability is a major concern for current MARL methods; however, I am a bit troubled by the lack of description in section 4.2. I am not very familiar with the cited meta-gradient method, and upon reading that paper, it is not clear how what is described in section 4.2 relates to that work. As such, I hope the authors can provide a more thorough explanation of this section in the rebuttal, such that I can evaluate their work more effectively.

[1] Samvelyan, Mikayel, et al. "The starcraft multi-agent challenge." arXiv preprint arXiv:1902.04043 (2019).

[2] Wang, Jianhao, et al. "QPLEX: Duplex Dueling Multi-Agent Q-Learning." International Conference on Learning Representations. 2020.

[3] Iqbal, S., Witt, C. A. S. D., Peng, B., Böhmer, W., Whiteson, S., & Sha, F. Randomized Entity-wise Factorization for Multi-Agent Reinforcement Learning. In Proceedings of the 38th International Conference on Machine Learning, ICML 2021

**Time Spent Reviewing:**

3

---

> ### Author Response · Authors · 2021-08-10
> **Response to Reviewer t76M**
>
> Thank you for the useful critique and the noted weaknesses of our paper as well as recommending REFIL as non-linear alternative to approximate sub-team values.
>
> **Section 4.2 (Meta-Gradient Learning)**
>
> The original approach was used to tune $\gamma$ and $\lambda$ of the TD($\lambda$)-objective using a straightforward gradient approach (Eq. 5 in the cited paper).
>
> We adopt this approach (Eq. 6 in our paper) in a novel context to tune sub-team assignments. For that, we define the optimization of agent sub-team assignments as a „sub-team selection“ problem or meta-policy ($\chi$) search. Thus, our meta-gradient resembles a standard policy gradient. While the tunable hyperparameters are different from the cited paper, the actual algorithm to tune them is the same.
>
> As a meta-learning approach, $\chi$ is updated in an outer loop (Line 18) of the actual training loop (Line 17) according to Algorithm 1 in our paper. This outer loop is necessary to evaluate the performance of the MAS separately before updating the meta-policy $\chi$ to avoid unintentional interference with the MARL process. Other meta-learning approaches like population-based training also update the meta-policy in an outer loop.
>
> **Evaluation and Scalability**
>
> Thank you for recommending QPLEX as alternative baseline. We see that the QPLEX has been officially presented at ICLR 2021 which was about 3 weeks before the NeurIPS deadline. However, we have noticed interesting connecting points while implementing QPLEX within the rebuttal phase, which we would like to discuss in Background, Related Work, and Discussion:
> 1. QPLEX defines another VFF operator that could be used as $\psi$ for VAST.
> 2. VAST could be reformulated for advantage-based IGM consistency by defining a dueling hierarchy, which propagates agent and sub-team advantages.
> 3. Besides REFIL, QPLEX itself is another potential candidate for non-linear sub-team value approximation because the dueling mixing component consists of two sums (the sum of Q-values and a weighted sum of advantages) which could be applied to variable team sizes.
> 4. Using QPLEX for $\psi$ and sub-team value approximation as proposed in 1. and 2. respectively could enable multi-level hierarchies of arbitrary depth and width.
>
> If $N$ is too small like, e.g., 4 agents in Warehouse or 20 agents in Battle, the additional overhead caused by the sub-teams in VAST might be rather prohibitive than helpful. Our results in Fig. 4 indicate that VAST performs especially well when $N$ is large (e.g., up to 80 or 800 agents).
> Since VAST builds upon existing VFF operators $\psi$ like QPLEX, QTRAN, QMIX, etc., we would expect the performance to be comparable to the original VFF operator or better for sufficiently large $N$. However, we see your point regarding StarCraft Multi-Agent Challenge (SMAC) and consider running a comparison for SMAC scenarios with $N > 20$, e.g., `25m`, `27m_vs_30m`, or `bane_vs_bane`, if our computational resources are sufficient.
>
> **Implementation Details (Lines 207-209)**
>
> Prior works often use very different $\epsilon$ annealing schedules, e.g., for QMIX, QPLEX, QTRAN which range from 50k to 3 million time steps depending on the domain and algorithms. This makes fair comparison and tuning for multiple domains and algorithms difficult.
> While we agree that actor-critic algorithms do not necessarily lead to better exploration, we just wanted to point out that we not tune any exploration parameter (e.g., entropy, noise, etc.) in particular. We simply used policy networks with the same structure and learning rate as the Q-networks which worked out of the box for all implemented algorithms and all domains. Thus, we could evaluate all approaches on a common basis - without further tuning.

---

### Official Review · Reviewer_2c76 · 2021-07-16

**Rating:** 6
**Confidence:** 3

**Summary:**

This paper proposes a novel value factorization method for multi-agent Q-learning based on team organization.
To enable automatic self-organization, this paper presents a meta-gradient-based approach to explore team structures.
The experiments demonstrate the proposed method can outperform baseline value factorization structures.

**Limitations And Societal Impact:**

I am willing to increase my score if my concerns are addressed. (1) and (2) are my major concerns, and (3) is minor.

(1) From my perspective, the concept of team coordination is a little bit weak in VAST. In general, VAST represents a subset of IGM factorization which focuses on individual execution. The selection of teams seems to only impact centralized training. In addition, I think it would be better to discuss some related work of coordination, such as deep coordination graph [1], and remark the differences from this paper.

(2) I am not sure whether I understand section 4.2 correctly. The proposed meta-gradient-based approach seems to require on-policy training. Whether this component will limit the sample efficiency and scalability?

(3) It is known that QTRAN may become unscalable in hard tasks, such as the StarCraft benchmark. It would be better to consider some more advanced techniques [2,3] and conduct more complex environments.

*References*

[1] Böhmer et al., Deep Coordination Graphs, ICML 2020.

[2] Rashid et al., Weighted qmix: Expanding monotonic value function factorisation, NeurIPS 2020.

[3] Wang et al., Qplex: Duplex dueling multi-agent q-learning, ICLR 2021.

**Main Review:**

**Originality**: VAST is well embedded in the widely-used CTDE  paradigm, and its novelty is clearly discussed. To my knowledge, the proposed factorization method is novel.

**Quality and Clarity**: The paper is well organized and easy to understand. Especially, Figure 1 is very clear and helpful to understand the proposed factorization structure.

**Significance**: Overall, this paper is a good start and attempt to study team coordination in multi-agent reinforcement learning. However, I have a few technical concerns (see questions).

**Time Spent Reviewing:**

4 hours

---

> ### Author Response · Authors · 2021-08-10
> **Response to Reviewer 2c76**
>
> Thank you for your positive review with important concerns and recommendation of relevant literature. We address your concerns as follows:
>
> **(1) Deep Coordination Graphs (DCG)**
>
> The motivation of DCG in [1] is similar to VAST, where the MAS is structured as a (connected) graph instead of disjoint sub-teams which is then exploited for VFF. Basically, DCG enriches VFF with agent relationship information, while VAST simplifies VFF via agent abstraction.
> - In contrast to DCG, the VAST team structure can vary between time steps depending on the assignment strategy and situation as demonstrated in Fig. 6. The paper [1] claims the possibility of generalization to different topologies but does not further evaluate this aspect. DCG focuses on pairwise agent interaction, while VAST has no restrictions regarding sub-team sizes over time.
> - DCG depends on a (flawless) communication channel for all agents to act, since their value functions depend on the neighbors' actions. In VAST, the local value functions are independent but consistent w.r.t. IGM due to the strict hierarchical scheme which we ensure during centralized training. Thus, all agents can act completely decentralized without iterative communication.
> - The maximization of $Q_{\textit{tot}}$ scales linearly for VAST w.r.t. $N$, while DCG scales at least quadratically if the graph is connected due to iterative communication. The evaluation of DCG in [1] is limited to 8 agents, while our experiments scale up to 16, 80, and 800 agents respectively, indicating the practical scalability of VAST w.r.t. $N$.
>
> **(2) On-Policy Meta-Gradient Learning**
>
> The current formulation of our meta-gradient approach according to Eq. 6 requires on-policy training and could indeed limit the sample efficiency especially regarding larger values of $\eta$ as observed in Fig. 4.
> However, this is an exemplary suggestion on how to implement an adaptive assignment strategy. While this already works well compared to alternative strategies as shown in Fig. 5, the meta-gradient approach could be further improved by using more sample efficient policy gradient variants like PPO, DDPG, or IMPALA. We deferred such an analysis to future work to keep the setup simple and to maintain focus on our main concept regarding sub-team based VFF.
>
> **(3) Advanced MARL Techniques**
>
> We agree, since QTRAN only performs well in the smaller domain instances. The scalability weakness of QTRAN further emphasizes the benefit of VAST in our experiments because VAST performs especially well with $\psi_{\textit{QTRAN}}$.
> Thank you for recommending two more recent techniques. Since WQMIX and QPLEX have been published very recently, we did not integrate these approaches for comparison yet. However, we find that both approaches are fairly easy to implement: WQMIX requires our QMIX implementation to integrate an additional Q-network and a loss weighting function and QPLEX requires a new module to compute $Q_{\textit{tot}}$ from $\langle Q_i \rangle_{i \in \mathcal{D}}$ ($V_i$ and $A_i$ are directly obtained from $Q_i$, therefore they can be encapsuled without affecting our training pipeline). In any case, we will include these techniques in Background, Related Work, and Discussion for completion.

---

### Official Review · Reviewer_pyHL · 2021-07-16

**Rating:** 7
**Confidence:** 3

**Summary:**

The paper introduces VAST, a framework to train decentralised MARL agents to each learn local policies from a single global reward. In contrast to VFF, where each agent calculates a local Q function which is then aggregated in a flat way into a total Q function to which Deep Q-Learning may be applied, VAST first assigns agents into variable-agent sub-teams, linearly aggregates the local Q function of each sub-team into a group Q function, and then aggregates the group function into a total Q function in the manner of VFF. The authors hypothesise that VFF underperforms with large numbers of agents due to a performance bottleneck that limits the information received by each agent, and empirically demonstrate a range of experiments where VAST achieves superior performance to flat versions of VFF when training large numbers of agents. They further discuss different ways of assigning agents to sub-teams, which may vary between time steps, and introduce an effective meta-gradient based approach.

**Limitations And Societal Impact:**

The authors discuss and acknowledge a range of possible negative societal problems, including use in autonomous weapons and traffic jams. I consider the most significant possible negative societal impact to be use in autonomous weapons, and would want to see a more detailed discussion of this. Eg, how often is MARL used today in autonomous weapons, and how much could this work further enable that?

**Main Review:**

Originality: This is a good, original work. It provides a novel variant of existing VFF techniques, building upon previous work while citing clearly.

Quality: It is technically sound, and the performance of VAST is well-supported with clear experimental results.

A major limitation is that there is insufficient discussion of the group ratio, and how this might be chosen. A key contribution of the work is the ability to scale VFF to larger numbers of agents, but the smallest group ratio considered is ¼, only a 4x decrease in the number of entities considered. It is likely that in settings with many agents we would want a significantly larger decrease. Eg, it would be instructive to see the case with just 2 or 3 sub-teams explored.

Clarity: The work is overall fairly well-written, well-organised and easy to follow. In particular, the background section was helpful and informative for putting the paper’s contribution into context. The experiments were well presented and clearly explained. A few points were unclear, as discussed below.

Significance: VAST outperforms previous approaches to VFF on MARL problems with large numbers of agents, a valuable contribution to an important problem.

The theoretical basis given for this work is shaky. The relative performance increases with larger numbers of agents, providing suggestive but not conclusive evidence for the hypothesis that this is due to a performance bottleneck. The authors should give more consideration to alternative explanations. Nevertheless, the empirical evidence for VAST is clear and persuasive.

Detailed comments:
- In 4.1 the ideal group ratio is described as $\frac{1}{N} < \eta \ll N$. As $\eta < 1 < N$, this is unclear, and $N$ should be replaced with $1$
- The phrase “performance bottleneck” feels misleading, and when reading the abstract I interpreted the authors as arguing that VFF was computationally intractable with large numbers of agents, and that VAST would resolve this.
- In 4.2 the discussion of the meta-gradients method for assigning teams was confusing, and could do with further detail, especially as we see that team assignment has a significant effect on performance. In particular: What does J mean, and how was this chosen? What are other reasonable choices of J, and how might this affect performance? How is $R_t$ included in calculations when this depends on the unknown future reward?
- Theorem 1 seems correctly proven, but is rigorously stating and proving the simple statement that “as team Q functions are linear aggregations of agent Q functions, they are maximised iff agent Q functions are maximised, so the total Q function is maximised iff each agent Q function is maximised”. To aid in reader understanding, the theorem should be accompanied by a more intuitive description.
- In Figure 6a-6c, $K=20$ yet only 2-4 groups are shown. The paper should clarify whether sub-teams are non-empty, and the impact on train time and performance of having many sub-teams with most empty vs having few sub-teams all non-empty.


**Time Spent Reviewing:**

7

---

> ### Author Response · Authors · 2021-08-10
> **Response to Reviewer pyHL**
>
> Thank you for your highly positive review and helpful comments:
>
> **Group Ratio $\eta$**
>
> On the current scale of our experiments, the difference between $\frac{1}{8}$ and $\frac{1}{4}$ would have been less significant for any conclusions or hypotheses (in the larger domain instances) than $\frac{1}{4}$ and $\frac{1}{2}$ which is why we stuck with this setting. $\frac{1}{4}$ was also the smallest possible value for $\eta$ in Warehouse[4], thus we kept it throughout the experiments for simplicity. However, we agree that a further evaluation on smaller group ratios in settings with much more agents would be helpful.
>
> **Detailed Comments**
>
> - Thank you for noting the typo.
> - In our view, the flat VFF structure limits performance in large MAS due to the limited representational capacity of the VFF operator $\psi$. Therefore, we could replace the term "bottleneck" by "(representational) capacity limit".
> - $J$ is the high-level objective function for optimizing sub-teams, which can be either domain dependent (e.g., the number of completed orders in Warehouse) or simply $R_t$, which we used to avoid additional domain dependencies (in contrast to $\chi_{Fixed}$ and $\chi_{Spatial}$). Optimizing sub-teams on an adequate high-level objective can guide the training process of the MAS to further improve performance as seen in the experiments. Since $\chi$ is updated at the end of each episode (line 18 in Algorithm 1), all rewards are available for computing $R_t$.
> - We appreciate your suggestion and consider adding more descriptive explanations by referring to Fig. 1b to visualize the intuition behind the theorem that the IGM consistency is still maintained through the hierarchy from agent to sub-teams and from sub-teams to $Q_{\textit{tot}}$ (given that $\psi$ maintains IGM consistency itself).
> - Although the number of non-empty sub-teams might be smaller than $K$ at each time step, the runtime is not affected because $\chi$ is called for each agent regardless of the number of non-empty sub-teams and $Q_{\textit{tot}}$ is always computed based on a fixed $K$ as input dimension even when most input entries are 0. Thus, the runtime still scales proportionally to $\eta$. We observed that $\chi_{MetaGrad}$ learns to use most sub-teams for very specific situations over time - but not necessarily all at once in the same situation as you correctly noted for Fig. 6a-c. This could be a hint that a much smaller $\eta$ of about $\frac{1}{20} (\ll 1/8 < 1/4)$ might be sufficient for Battle[80].
>
> **Negative Societal Impact: Autonomous Weapons**
>
> Given the high sensitivity of this topic and the lack of publicly available information, we can only speculate about that. Autonomous weapon systems are a highly potential application field for RL and MARL, due to the recent progress in robotics and their ability to learn unconventional strategies which are hard to predict and to counter by humans as shown in AlphaGo, OpenAI Five or AlphaStar. Since current state-of-the-art MARL just scales to a few agents, this would only suffice to control a handful of units (e.g., drones), thus limiting MARL for real world use. However, VAST enables MARL to scale up to much more units and potentially offering more resilience against single unit failures as observed in the Battle domain, where single agents can be killed and removed from the game.

---

### Official Review · Reviewer_bA39 · 2021-07-19

**Rating:** 6
**Confidence:** 5

**Summary:**

The paper takes a hierarchical approach rather than the now fairly common flat approach to factorizing joint value function for multiple agents in multi-agent reinforcement learning. Sub-team assignments can change for agents over time. Additionally, the paper explores different sub-team assignment strategies and proposes a meta-gradient based approach to optiize these assignments. The proposed method is compared against baselines that have a flat factorization on multiple grid-based domains and shown to outperform them especially as with increasing number of agents.

**Limitations And Societal Impact:**

The paper goes in quite a bit detail in Sec. 8 for potential societal impact. As mentioned above an important thread from the literature is missing but otherwise the limitations are interspersed through the text.

**Main Review:**

**Strengths**

The paper is clearly written and the experiments are well done with results reported over 30 runs. Moreover the experiments go to about 80 coordinating agents unlike most other works in this domain. Unlike many papers, the delta difference at the end of training is quite substantial as the number of agents increases. Ablations with respect to different assignment strategies are quite useful even though they don't seem to matter always. The shared source code is fairly easy to follow as well, so kudos for that.

**Weaknesses**

Biggest weakness is the paper's lack of recognition of alternative value function factorizations based on ideas of coordination graphs [1,2,3]. These are not exactly flat, but can't be claimed to be hierarchical in this manner. But given the paper uses a different set of task domains than the other papers, it's difficult to infer the pros and cons of these different approaches. Nevertheless, at least some discussion is warranted. From the code it seems the assignment learning is another stochastic optimization problem that would likely become difficult with larger number of sub-teams and is not _just_ about linear scaling. That is possibly what is happening with Battle[80]. Some more discussion would be useful here. In general, the paper reports a bunch of results but doesn't try to figure out or given any intuition for why it could be happening. I understand that this is difficult and often not feasible with deep neural network based methods, but still some hypotheses would be useful!

[1] W Böhmer et al. "Deep Coordination Graphs" https://arxiv.org/abs/1910.00091

[2] S Li et al. "Deep Implicit Coordination Graphs for Multi-agent Reinforcement Learning" https://arxiv.org/abs/2006.11438

[3] N Naderializadeh et al. "Graph Convolutional Value Decomposition in Multi-Agent Reinforcement Learning" https://arxiv.org/abs/2010.04740

**Time Spent Reviewing:**

3

---

> ### Author Response · Authors · 2021-08-10
> **Response to Reviewer bA39**
>
> Thank you for your positive review and feedback regarding highly related work:
>
> **Coordination Graphs (CG)**
>
> The motivation of CG is similar to VAST, where the MAS is structured as a (connected) graph instead of disjoint sub-teams which is then exploited for VFF. Basically, CG enriches VFF with agent relationship information, while VAST simplifies VFF via agent abstraction.
> - CG focuses on the interaction between a fixed number of agents (typically pairwise), while VAST has no restrictions regarding sub-team sizes over time. Thus, VAST models agent-to-sub-team relationships via sub-team assignment operators rather than agent-to-agent relationships via edges in a graph.
> - Most CG approaches depend on a (flawless) communication channel for all agents to act, since their value functions depend on the neighbors' actions. In VAST, the local value functions are independent but consistent w.r.t. IGM due to the strict hierarchical scheme which we ensure during centralized training. Thus, all agents can act completely decentralized without iterative communication.
> - The maximization of $Q_{\textit{tot}}$ scales linearly for VAST w.r.t. $N$, while CG approaches scale at least quadratically if the graph is connected either due to iterative communication [1] or centralized training [2,3], where the full graph structure (e.g., adjacency matrices in GCN) is processed.
>
> **Assignment Learning in our Code**
>
> The computational scaling is linear w.r.t. $N$ for meta-gradient assignment learning, since $\chi$ is invoked for each agent separately and trained on a data amount which is proportional to $N$.
> We agree that the difficulty of finding good solutions increases with the number of sub-teams, which is why we recommend a small $\eta$ for large $N$. We assume that our meta-gradient approach lacks sufficient exploration to search in large (sub-team) spaces which could be further improved by, e.g., adding an entropy bonus or exploration noise.

---

### Decision · Program_Chairs · 2021-09-27

**Decision:**

Accept (Poster)

**Comment:**

The reviewers in most part recognise the novelty/originality of the presented approach.  The experiments, through which a strong  case is made for the scalability of the method to large number of agents, seem solid and competitive with state of the art. Some concerns raised by  Reviewer t76M about the clarity of section 4.2  on metagradient subroutine which needs to be addressed in the final version.